# CAN: Leveraging Clients As Navigators for Generative Replay in Federated Continual Learning

Xuankun Rong [* 1]   Jianshu Zhang [* 2]   Kun He [2]   Mang Ye [1]

## Abstract

Generative replay (GR) has been extensively validated in continual learning as a mechanism to synthesize data and replay past knowledge to mitigate forgetting. By leveraging synthetic rather than real data for the replay, GR has been adopted in some federated continual learning (FCL) approaches to ensure the privacy of client-side data. While existing GR-based FCL approaches have introduced improvements, none of their enhancements specifically take into account the unique characteristics of federated learning settings. *Beyond privacy constraints, what other fundamental aspects of federated learning should be explored in the context of FCL?* In this work, we explore the potential benefits that come from emphasizing the role of clients throughout the process. We begin by highlighting two key observations: (a) Client Expertise Superiority, where clients, rather than the server, act as domain experts, and (b) Client Forgetting Variance, where heterogeneous data distributions across clients lead to varying levels of forgetting. Building on these insights, we propose **CAN** (**C**lients **A**s **N**avigators), highlighting the pivotal role of clients in both data synthesis and data replay. Extensive evaluations demonstrate that this client-centric approach achieves state-of-the-art performance. Notably, it requires a smaller buffer size, reducing storage overhead and enhancing computational efficiency.

---

[*]Equal contribution  [1]National Engineering Research Center for Multimedia Software, School of Computer Science, Wuhan University, Wuhan, China [2]School of Cyber Science and Engineering, Wuhan University, Wuhan, China. Correspondence to: Mang Ye <yemang@whu.edu.cn>.

*Proceedings of the 42^nd International Conference on Machine Learning*, Vancouver, Canada. PMLR 267, 2025. Copyright 2025 by the author(s).

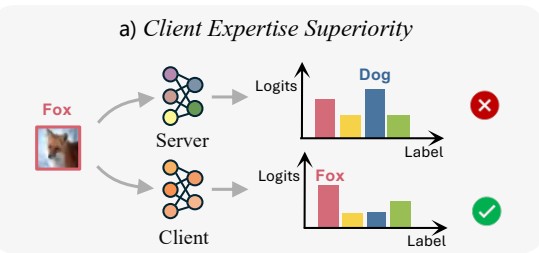

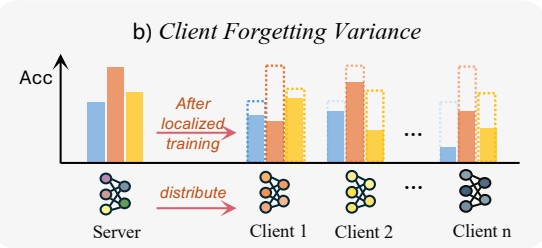

*Figure 1.* a) *Client Expertise Superiority* (Sec. 3.1): Clients may possess more accurate knowledge of specific classes compared to the server, highlighting the need to utilize client expertise during synthetic data generation. b) *Client Forgetting Variance* (Sec. 3.2): The non-IID nature of client data leads to diverse forgetting patterns across clients, emphasizing the need for adaptive replay strategies tailored to each client.

## 1. Introduction

Federated Learning (FL) (Konečný, 2016; McMahan et al., 2017; Yang et al., 2019; Li et al., 2020a; Huang et al., 2024; 2023a) is a learning paradigm that utilizes distributed data across clients rather than relying on centralized datasets. Clients train models locally on their own data, which are then aggregated into a global model on the server (Li et al., 2020b; Karimireddy et al., 2020; Fallah et al., 2020; Pi et al., 2023; Xie et al., 2022). However, one key challenge in this paradigm is that data distributions can change over time (Liu et al., 2024; Fang et al., 2023; Huang et al., 2023b; Gao et al., 2022a; Jiang et al., 2022). For example, user preferences in mobile apps might change (Ma et al., 2022), sensor data collected by IoT devices could vary with environmental conditions (Nguyen et al., 2021), or healthcare records may evolve as new medical trends emerge (Xu et al., 2021; Wang et al., 2023). These evolving distributions can lead to catastrophic forgetting, a challenge that is particu-

larly pronounced in the distributed nature of FL.

To address forgetting, recent studies (Dong et al., 2022; Ma et al., 2022; Dai et al., 2024; Yang et al., 2024; Usmanova et al., 2021) have introduced Federated Continual Learning (FCL), adapting traditional Continual Learning (CL) methods to the FL setting (Huszár, 2018; Aljundi et al., 2019; Zhao et al., 2020; Zhou et al., 2023). To address privacy concerns in FL, most approaches naturally adopt generative replay (Zhang et al., 2023; Tran et al., 2024; Mei et al., 2024; Liang et al., 2025), which leverages a generator to synthesize data for replay, effectively mitigating forgetting. However, we observed that current approaches often make compromises to meet strict privacy constraints. This raises an important question: *Beyond privacy constraints, what other fundamental aspects of federated learning should be explored in the context of FCL?*

To fill this gap in the current FCL field, we shift our focus to a crucial feature of FL: **Clients** (Yang et al., 2023; Shanmugarasa et al., 2023; Fu et al., 2023; Fang & Ye, 2022; Yu et al., 2024b; Ma et al., 2025). By exploring the unique characteristics that clients can contribute, we aim to help FCL better leverage the distinctive features of the FL scenario. As illustrated in Fig. 1, we explore two key observations that have been largely overlooked in FCL approaches: ❶ *Client Expertise Superiority*: Due to the non-IID nature of client data in FL settings (Zhao et al., 2018; Konečný et al., 2016; Huang et al., 2022; Ye et al., 2023), some clients naturally become domain experts in specific categories, especially as the degree of non-IID increases (as visualized in Fig. 2). Although these clients may not perform as well as the server in a general sense, they often possess more precise and specialized knowledge in particular areas. This makes them especially valuable for guiding the generation of high-quality synthetic data in those domains. However, current approaches focus solely on the server model's knowledge, missing the opportunity to leverage this client-specific expertise. ❷ *Client Forgetting Variance*: Due to clients' unique local data distributions (Mendieta et al., 2022; Singhal et al., 2021; Yu et al., 2024a), they exhibit varying forgetting patterns. Despite this, existing methods use a uniform replay buffer, where each client receives an identical buffer with an even distribution of synthetic data samples across classes. This uniform allocation fails to meet the unique needs of each client, who may benefit more from a tailored data replay strategy to mitigate forgetting.

Based on the two observations above, we propose **CAN** (**C**lients **A**s **N**avigators), a more native FCL approach that leverages the unique role of clients. Our method follows the basic generative replay framework by first utilizing the teacher model's knowledge to train a generator for synthesizing data (Zhang et al., 2023; Tran et al., 2024; Shin et al., 2017; Wang et al., 2024b). The synthetic data is then com-

bined with newly received data for clients' local training. However, unlike previous methods, CAN takes a *client-centric* approach, where clients navigate the generation of synthetic data and guide targeted replay.

Specifically, our approach can be divided into two stages. The first stage, *Expert-Driven Data Synthesis*, focuses on generating high-quality synthetic data. For the first time, we position clients as experts, leveraging their specialized knowledge to guide the generator in producing accurate synthetic data. Additionally, to prevent synthetic data from overemphasizing only the most prominent features, we encourage the generator to capture more subtle characteristics—those discernible to domain experts but often overlooked by less specialized models. This allows the synthetic data to be both finely detailed and comprehensive. The second stage is *Adaptive Replay*. It starts with measuring the unique forgetting patterns of each client, influenced by the varying degrees of interference between newly acquired knowledge and prior knowledge. This process enables us to create a distinct forgetting profile for each client. Next, we adjust the buffer's data composition according to each client's forgetting profile, tailoring the allocation of data within the buffer. By differentiating the buffer across clients, we enable a more targeted and effective replay process. Our contributions can be summarized as follows:

- We identify that existing FCL methods fall short in fully leveraging the unique characteristics of FL. For the first time, we introduce a client-centric perspective and propose two key observations that highlight the necessity of positioning clients as navigators in FCL.

- We propose CAN that leverages the unique role of clients in FL. By positioning clients as navigators to guide the generation of synthetic data and adaptive replay tailored to each client's needs, CAN enables more efficient and targeted knowledge retention.

- Through extensive experiments on CIFAR100, TinyImagenet, and Imagenet100, we achieve state-of-the-art (SOTA) performance, with ablation studies validating the effectiveness of our client-centric approach.

## 2. Related Work

**Continual Learning.** Methods in continual learning can be broadly categorized into three approaches (De Lange et al., 2021; Wang et al., 2024a). Parameter isolation methods (Yoon et al., 2017), such as Progressive Neural Networks (PNN (Rusu et al., 2016)) and context-dependent gating (XdG (Masse et al., 2018)), segregate parameters to minimize task interference but face scalability issues as knowledge expands. Regularization approaches, including Elastic Weight Consolidation (EWC (Huszár, 2018)) and

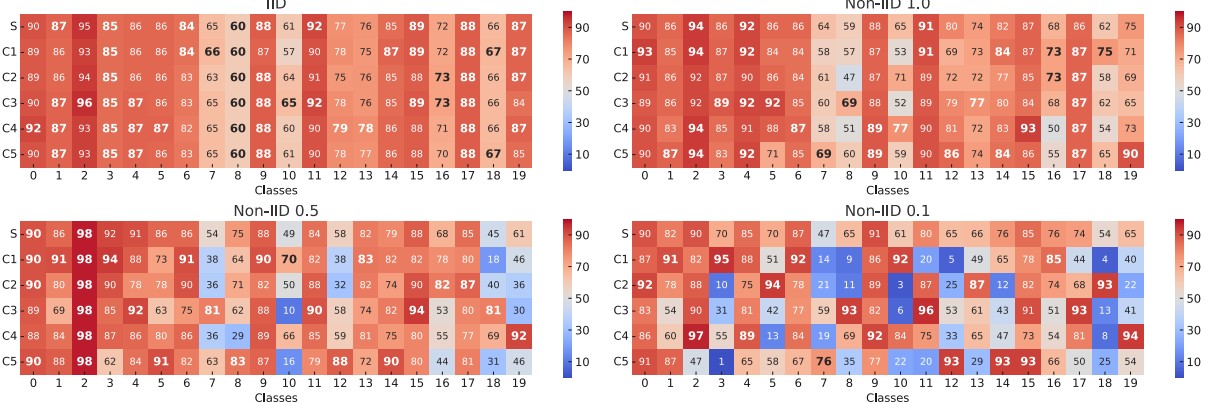

*Figure 2.* This visualization shows accuracy across different classes under IID and Non-IID (0.1, 0.5, 1.0) settings. The server (S) and clients (C1 to C5) are compared, with the highest accuracy for each class highlighted in **bold**. Certain clients outperform the server on specific classes, demonstrating their potential as class-specific experts. This observation supports the strategy of leveraging client expertise to improve performance on data synthesis. Refer to Sec. 3.1 for details.

Learning Without Forgetting (LwF (Li & Hoiem, 2017)), protect previously acquired knowledge by limiting parameter updates. However, their effectiveness depends heavily on accurately identifying important parameters. Data replay methods (Shin et al., 2017; Zhai et al., 2019; Masarczyk & Tautkute, 2020), like Cognitive Replay (CORE (Zhang et al., 2024)), Experience Replay (ER (Rolnick et al., 2019)) and Incremental Classifier and Representation Learning (iCaRL (Rebuffi et al., 2017)), use a replay buffer to review historical data.

**Federated Continual Learning.** Federated Continual Learning (FCL) (Bui et al., 2018; Dong et al., 2022; Ma et al., 2022; Yoon et al., 2021; Gao et al., 2024; Yu et al., 2024c) extends Continual Learning (CL) by integrating Federated Learning (FL) principles (Grammenos et al., 2020; Benmalek et al., 2022; Li et al., 2021), allowing global models to adapt to evolving local data while preserving prior knowledge and ensuring privacy. Existing approaches can be grouped into several categories: 1) regularization-based methods like FedCurv (Shoham et al., 2019), which use penalty terms to preserve critical parameters; 2) knowledge distillation methods such as CFeD (Ma et al., 2022) and GLFC (Dong et al., 2022), which transfer essential knowledge between tasks to mitigate forgetting; 3) exemplar-based methods like FedWeIT (Yoon et al., 2021), which retain key samples for rehearsal during new task training; 4) generative replay-based methods, including TARGET (Zhang et al., 2023), LANDER (Tran et al., 2024), and DDDR (Liang et al., 2025; Mei et al., 2024), primarily build upon standard generative replay, with most improvements focusing on integrating pretrained models, such as CLIP and diffusion models, to enhance performance. In contrast, we re-examine the advantages that clients can offer in FL scenarios and propose a more FL-native approach.

# 3. Observations

Clients play a crucial role in FL scenarios, which brings two defining characteristics: 1) the privacy of client-side data, and 2) the non-IID (uneven) distribution of data across clients. In transitioning from CL to FCL, existing methods have mainly focused on ensuring data privacy by keeping local data within clients. However, they often overlook the non-IID distribution of client data. In this section, we shift our focus to this underutilized feature, discussing the potential benefits it can bring to FCL.

## 3.1. Client Expertise Superiority.

As illustrated in Fig. 2, under an IID data distribution, both clients and the server achieve comparable performance, with each class exhibiting relatively high accuracy. However, in practical FL scenarios, client data distributions often vary significantly, leading to inherently non-IID settings. We observe that as the level of non-IID increases, clients encounter an increasing number of low-accuracy instances. At the same time, certain clients also achieve notably higher accuracy on specific classes. This trend is more pronounced in the non-IID (0.1) setting, where a strong polarization effect emerges—individual clients demonstrate widely varying performance across different classes, with some significantly outperforming the server on specific categories.

Leveraging these high-performing clients as domain experts for certain classes offers a more effective strategy than relying solely on server knowledge. By incorporating their superior knowledge, we can enhance the generation of synthetic data, ensuring a better alignment with real-world FL scenarios. Furthermore, during server-side aggregation, these expert client models can be integrated without exposing local data, thereby maintaining privacy. Notably, this approach incurs minimal storage overhead, as it only

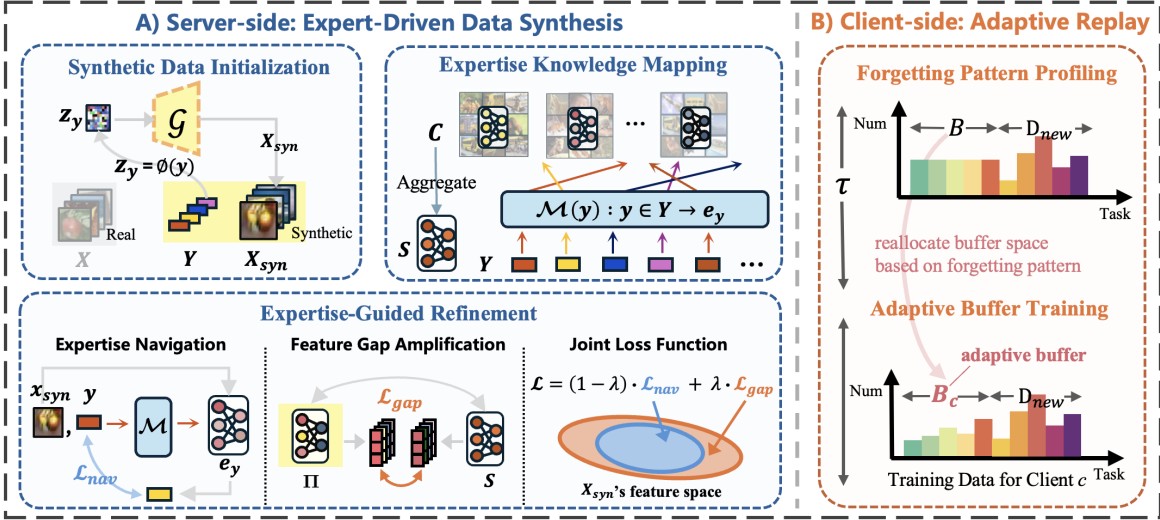

Figure 3. Overview of CAN, consisting of two main stages: Expert-Driven Data Synthesis (Sec. 4.2) and Adaptive Replay (Sec. 4.3). A), we start with initializing synthetic data. Then we identify the most suitable client experts to navigate the data generation. The process is further refined using a joint loss that ensures both accurate and detailed data. B), the replay process begins with analyzing how new data affects each client's retention of old knowledge. This profiling informs Adaptive Buffer Training, where we personalize the internal data distribution of replay buffers based on the unique forgetting patterns of each client, enabling more tailored and effective replay.

introduces an additional inference step into the existing aggregation process.

### 3.2. Client Forgetting Variance.

In FCL scenarios, the degree of interference with previously learned knowledge varies significantly across clients due to their diverse data distributions. Each client's unique local data introduces different challenges in balancing new and old knowledge, leading to varying degrees of conflict. For instance, some clients may encounter frequent overlaps between new and previously seen data, causing a higher risk of interference, while others, with less overlap, might experience distinct but subtle shifts that still lead to forgetting. This variability naturally results in different levels of interference and forgetting among clients.

Consequently, a one-size-fits-all replay strategy is insufficient to address these differences. Instead, replay buffers should be adaptive, tailored to each client's specific forgetting patterns. Recognizing and adapting to this client-specific variability is essential for effective FCL.

## 4. CAN: Clients As Navigators

### 4.1. Overview

In this section, we introduce **CAN**, a native FCL approach that leverages clients' roles in FL to guide replay. We treat clients as experts to train a generator that synthesizes fine-grained data, effectively preserving prior knowledge. Ad-

ditionally, we personalize each client's replay buffer based on their forgetting patterns, enhancing replay effectiveness. The framework of CAN is illustrated in Fig. 3.

### 4.2. Expert-Driven Data Synthesis

#### 4.2.1. SYNTHETIC DATA INITIALIZATION

Due to privacy concerns, the real data owned by clients is inaccessible on the server side. To overcome this limitation, we introduce a generator, denoted as $\mathcal{G}$, to produce synthetic data $X_{\text{syn}}$ that aims to approximate the real data $X$ in distribution. This enables data replay using synthetic data, preserving knowledge from previous tasks while addressing privacy concerns. Here, we first adopt a sampling function $\phi(y)$ that applies a noise transformation to obtain $z_y$. Then, the generator $\mathcal{G}$ takes noise $z_y$ as input to generate synthetic data $x_{\text{syn}}$ corresponding to that class. The process can be formulated as:

$$X_{\text{syn}} \sim \mathbb{P}_X = \{x_{\text{syn}} = \mathcal{G}(z_y) \mid z_y \leftarrow \phi(y),\ y \in \mathcal{Y}\},\ (1)$$

where $\mathbb{P}_X$ represents the distribution of the real data $X$, and $\phi(y)$ is a sampling function that generates the input noise $z_y$ for each label $y$. Additionally, we discuss later in Sec. 5.5, the synthetic data does not visually resemble the real data, but remains unrecognizable and serves as meta data.

#### 4.2.2. EXPERTISE KNOWLEDGE MAPPING

To improve the quality of $X_{\text{syn}}$, it is essential to have an effective guide for optimizing $\mathcal{G}$. In light of our observa-

tion of *Client Expertise Superiority* in Sec. 3.1, we stress the importance of leveraging the expertise of the clients. Specifically, for each class $y \in \mathcal{Y}$, we utilize an expertise map to identify the most suitable expert from clients $\mathcal{C}$. The expertise map $\mathcal{M}(\cdot)$ is defined as follows:

$$\mathcal{M}(y) : y \in \mathcal{Y} \mapsto e_y = \arg\max_{c \in \mathcal{C}} \text{Acc}(c, y), \qquad (2)$$

here, $\mathcal{M}(\cdot)$ selects the expert $e_y$ with the highest accuracy for class $y$. Once the generator $\mathcal{G}$ produces the synthetic data $(x_{\text{syn}}, y)$, the selected expert $e_y$ is then used to guide the subsequent data refinement process.

### 4.2.3. EXPERTISE-GUIDED REFINEMENT

In this part, our goal is to refine synthetic data to not only capture the core characteristics of the original data but also encompass diverse and nuanced features. To achieve this, we implement a cooperative refinement process with two key components: *Expertise Navigation*, which ensures accurate and class-specific representation, and *Gap-Driven Enhancement*, which promotes richer details.

**Expertise Navigation.**   To ensure that the synthetic data accurately represents the characteristics of its corresponding class, we adopt a strategy that treats a well-trained model as an expert. This expert facilitates knowledge transfer and guides the generation of representative synthetic data. However, unlike previous approaches that rely solely on the server as the expert, our method takes full advantage of the FL setting by positioning clients as experts. This allows us to harness the unique strengths of different clients, who may have a deeper understanding of particular data distributions compared to the server.

We first use the expertise map $\mathcal{M}(\cdot)$ to select the most suitable expert for each class, ensuring more accurate and targeted refinement. The selected expert $e_y$ then generates a predicted label for the synthetic data $x_{\text{syn}}$, corresponding to class $y$. Given that this expert is highly proficient in recognizing its associated class, the predicted label serves as a reliable proxy for the ground truth, denoted as $y_{\text{gt}}$.

When the synthetic data accurately captures the characteristics of class $y$, the expert model $e_y$ should be able to classify it correctly. Therefore, we compute a cross-entropy loss between the predicted class $y_{\text{gt}}$ from the expert and the target class $y$, formulated as follows:

$$\mathcal{L}_{\text{nav}} = -\log p(y_{\text{gt}} = y | x_{\text{syn}}), \qquad (3)$$

through $\mathcal{L}_{\text{nav}}$, the generator $\mathcal{G}$ can be optimized to generate synthetic data that effectively represents the target class.

**Feature Gap Amplification.**   It is important to note that the loss function $\mathcal{L}_{\text{nav}}$ in Eq. (3) only requires the synthetic data to be identifiable by the expert, but it does not guarantee that all relevant features are captured. Specifically, the generator tends to focus on the most prominent and easily recognizable features to satisfy the expert, while ignoring the more nuanced and detailed characteristics that the expert is also capable of recognizing. Consequently, the generated data often lack subtle but important features, resulting in reduced diversity and limiting their representational richness.

To address this, we introduce a reference model $\Pi$, trained only on synthetic data that can be understood as a knowledge representation under the guidance of expert knowledge. Therefore, $\Pi$ captures not only the prominent features but also includes some of the more subtle and nuanced characteristics that are often overlooked by less specialized models. In contrast, the server model $\mathcal{S}$, while proficient at identifying prominent features, lacks the expert's capacity to capture these finer ones. By encouraging discrepancies between the predictions of the server model $\mathcal{S}$ and the reference model $\Pi$ on $\mathcal{X}_{\text{syn}}$, we guide the generator $\mathcal{G}$ to produce synthetic data that not only captures the prominent features but also the fine-grained details. The features recognized by the reference model $\Pi$ can be represented as a set $F_\Pi(x)$, and those recognized by the server model $\mathcal{S}$ as a set $F_\mathcal{S}(x)$.

Our objective is to let synthetic data capture the features recognized by $\Pi$ but missed by $\mathcal{S}$, which can be understood as the set difference $F_\Pi(x) \setminus F_\mathcal{S}(x)$ in discrete mathematics theory. To achieve this, we aim to minimize the KL divergence (Van Erven & Harremos, 2014) between the predictions of $\Pi$ and $\mathcal{S}$ on the test set, focusing only on cases where $\Pi$ makes correct predictions. The process is formalized as:

$$\mathcal{L}_{\text{gap}} = -\mathbb{E}_{(x,y)\sim\mathcal{X}_{\text{syn}}} \left[ \mathbb{I}_\Pi \cdot \text{KL}(f_\Pi(x) \| f_\mathcal{S}(x)) \right], \qquad (4)$$

where $\mathbb{E}_{(x,y)\sim\mathcal{X}_{\text{syn}}}$ denotes the expectation over $\mathcal{X}_{\text{syn}}$. Here, $f_\Pi(x)$ and $f_\mathcal{S}(x)$ represent the predicted label by $\Pi$ and $\mathcal{S}$, respectively. The indicator function $\mathbb{I}_\Pi$ equals 1 if $\Pi$'s prediction matches the true label $y$, and 0 otherwise, ensuring that discrepancies are encouraged only when $\Pi$ correctly identifies the features.

**Joint Loss Function.**   To comprehensively optimize the generator $G$, we define a joint loss function that combines both $\mathcal{L}_{\text{nav}}$ and $\mathcal{L}_{\text{gap}}$. The navigation loss $\mathcal{L}_{\text{nav}}$ aligns the synthetic data with the expert's guidance to accurately represent the target class, while the gap-driven loss $\mathcal{L}_{\text{gap}}$ promotes diversity by encouraging the capture of more nuanced details. The overall objective for optimizing $G$:

$$\mathcal{L}_G = (1 - \lambda) \cdot \mathcal{L}_{\text{nav}} + \lambda \cdot \mathcal{L}_{\text{gap}}, \qquad (5)$$

here, $\lambda$ is a hyperparameter balances between the two loss terms, ensuring that the synthetic data is both representative and diverse. Further discussion of different $\lambda$ can be found in Appendix A.3.2.

## 4.3. Adaptive Replay

A common approach to mitigate catastrophic forgetting in continual learning is to mix previously data, representing old knowledge, with newly incoming data during training. However, in FL settings, distributed clients have highly heterogeneous local data, contributing the *Client Forgetting Variance* phenomenon discussed in Sec. 3.2. This makes naive replay without considering the unique role of clients unlikely to yield optimal results. To improve the effectiveness of replay in the FL scenario, we propose an adaptive replay strategy: 1) Forgetting Pattern Profiling, which profiles the unique forgetting patterns of each client; 2) Adaptive Buffer Training, which reallocate personalized buffer for each client during replay.

**Forgetting Pattern Profiling.** Initially, a replay buffer $\mathcal{B}$ is pre-allocated to all clients, with synthetic data evenly distributed across all previous tasks $\mathcal{T}$:

$$\mathcal{B} = \left\{ (\mathcal{X}_{\text{syn}}^t, \mathcal{Y}^t) \mid t \in \mathcal{T}, |\mathcal{X}_{\text{syn}}^t| = \frac{|\mathcal{B}|}{|\mathcal{T}|} \right\}, \quad (6)$$

where $|\cdot|$ denotes the size of a set. Each client then trains on the replay buffer $\mathcal{B}$ alongside the new local data $\mathcal{D}$ for $\tau$ iterations. The choice of $\tau$ controls the balance between efficient training and capturing the impact of the new task, as detailed in Appendix A.3.3.

After training, we evaluate the accuracy of each client $c$ on each old task $t$, denoted as $\text{Acc}_c^t$, to determine the forgetting pattern for each client. The forgetting weight $w_c^t$ for client $c$ on old task $t$ is calculated as follows:

$$w_c^t = \frac{1}{\max(\text{Acc}_c^t, \epsilon)}, \quad t \in \mathcal{T}, \quad (7)$$

where $\epsilon$ is a small predefined threshold to prevent numerical instability when the accuracy is close to zero. A larger $w_c^t$ indicates greater forgetting, suggesting that more data is needed for client $c$'sreplay on task $t$.

**Adaptive Buffer Training.** Next, we implement a simple adaptive strategy to allocate more buffer space to tasks with higher forgetting weights. The adaptive replay buffer $\mathcal{B}_c$ for client $c$ is defined as:

$$\mathcal{B}_c = \{(\mathcal{X}_{\text{syn}}^t, \mathcal{Y}^t) \mid t \in \mathcal{T}, |\mathcal{X}_{\text{syn}}^t| = \frac{w_c^t}{\sum_{t' \in \mathcal{T}} w_c^{t'}} \cdot |\mathcal{B}|\}, \quad (8)$$

where $w_c^t$ represents the forgetting weight for task $t$ on client $c$, and $|\mathcal{X}_{\text{syn}}^t|$ is the size of the buffer allocated to task $t$ after adaptation. Following this, each client $c$ trains on their personalized buffer $\mathcal{B}_c$ alongside their new local data $\mathcal{D}_c$. Finally, the loss of the client $c$ after task $t$ is defined as:

$$\mathcal{L}_c = \alpha_{new}^t \cdot \mathbb{E}_{\mathcal{D}_c}[\text{CE}] + \alpha_{pre}^t \cdot \mathbb{E}_{\mathcal{B}_c}[\text{KL}], \quad (9)$$

where $\alpha_{new}^t$ and $\alpha_{pre}^t$ balance learning from new data $\mathcal{D}_c$ and retaining past knowledge via buffered data $\mathcal{B}_c$. The specific selection of $\alpha$ is detailed in Appendix A.3.4.

# 5. Experiments

## 5.1. Experimental Setup

**Settings and Metrics.** We evaluate CAN in both IID and diverse non-IID scenarios using two key metrics: Average Accuracy ($\mathcal{A}$) and Average Forgetting Score ($\mathcal{F}$), as detailed in Appendix A.1. Additionally, we also examine the impact of different replay buffer sizes to assess the efficiency of our approach under resource constraints. Furthermore, to provide a comprehensive evaluation, we analyze accuracy trajectory across sequential tasks and retention of initial task accuracy, offering deeper insights into the model's capacity to maintain performance throughout training.

**Baselines.** We compare CAN with several methods: Vanilla, FedEWC, FedLwF, FedWeIT, TARGET, and LANDER. Vanilla sequentially finetunes each new class. FedEWC (Kirkpatrick et al., 2017) mitigates forgetting by penalizing changes to important parameters using elastic weight consolidation (EWC). FedLwF (Li & Hoiem, 2017) applies knowledge distillation to retain past knowledge. FedWeIT (Yoon et al., 2021) preserves crucial weights in federated settings. TARGET (Zhang et al., 2023) generates synthetic data for replay using the server model's knowledge, avoiding real data access. LANDER (Tran et al., 2024), an extension of TARGET, introduces feature anchors to improve synthetic data clustering. Additionally, we compare CAN with DDDR (Liang et al., 2025), which leverages a pretrained diffusion model (see Appendix B).

## 5.2. Main Results on CIFAR100

Tab. 1 compares various methods on CIFAR100 (Krizhevsky & Hinton, 2009) across IID and various non-IID settings, where our method consistently outperforms others across all settings. As expected, Vanilla sequentially fine-tunes new tasks, leading to severe forgetting. Among non-replay methods like FedEWC (Kirkpatrick et al., 2017), FedWeIT (Yoon et al., 2021), and FedLwF (Li & Hoiem, 2017), forgetting is somewhat alleviated, but these approaches struggle under non-IID conditions due to the absence of synthetic data replay. Replay-based methods like TARGET (Zhang et al., 2023) and LANDER (Tran et al., 2024), which serve as our main baselines, use synthetic data to replay. TARGET relies on server-side knowledge with a larger buffer but exhibits high forgetting in non-IID settings. LANDER introduces feature anchors to improve clustering, but it struggles to match our method's performance at the same buffer size. In fact, CAN outperforms LANDER even with smaller buffers, highlighting CAN's efficiency in terms of computational

*Table 1.* **Performance on CIFAR100** for 5-task and 10-task settings under IID and various non-IID scenarios. For TARGET, we replicate it using a buffer size of 2,560 while following its default settings. Our main comparison is with LANDER, exploring different buffer sizes: 1,250 , 1,000 , 750 , and 500 . The results demonstrate that CAN consistently outperforms LANDER at equivalent buffer sizes and achieves comparable performance even with smaller buffers, highlighting its efficiency and effectiveness in maintaining high accuracy ($\mathcal{A}$) and reducing forgetting score ($\mathcal{F}$). Further analysis can be found in Sec. 5.2.

| | 5 Tasks | | | | | | | | 10 Tasks | | | | | | | |
| :---: | :---: | :---: | :---: | :---: | :---: | :---: | :---: | :---: | :---: | :---: | :---: | :---: | :---: | :---: | :---: | :---: |
| **Method** | **IID** | | **NIID(1)** | | **NIID(0.5)** | | **NIID(0.1)** | | **IID** | | **NIID(1)** | | **NIID(0.5)** | | **NIID(0.1)** | |
| | $\mathcal{A}(\uparrow)$ | $\mathcal{F}(\downarrow)$ | $\mathcal{A}(\uparrow)$ | $\mathcal{F}(\downarrow)$ | $\mathcal{A}(\uparrow)$ | $\mathcal{F}(\downarrow)$ | $\mathcal{A}(\uparrow)$ | $\mathcal{F}(\downarrow)$ | $\mathcal{A}(\uparrow)$ | $\mathcal{F}(\downarrow)$ | $\mathcal{A}(\uparrow)$ | $\mathcal{F}(\downarrow)$ | $\mathcal{A}(\uparrow)$ | $\mathcal{F}(\downarrow)$ | $\mathcal{A}(\uparrow)$ | $\mathcal{F}(\downarrow)$ |
| Vanila | 16.12 | 78.12 | 16.33 | 77.59 | 15.49 | 74.95 | 15.69 | 72.73 | 7.83 | 75.89 | 8.45 | 74.89 | 7.64 | 71.52 | 8.04 | 65.27 |
| FedEWC | 16.51 | 71.12 | 16.06 | 68.02 | 16.86 | 62.40 | 17.69 | 65.84 | 8.01 | 65.06 | 8.84 | 62.14 | 8.04 | 65.23 | 7.61 | 58.79 |
| FedWeIT | 28.45 | 52.12 | 28.56 | 49.84 | 24.57 | 45.96 | - | - | 20.39 | 43.18 | 19.68 | 45.96 | 15.45 | 48.54 | - | - |
| FedLwF | 30.61 | 45.32 | 30.94 | 42.71 | 27.59 | 41.25 | 35.95 | 29.04 | 23.27 | 37.71 | 21.16 | 41.03 | 17.98 | 45.23 | 18.39 | 43.92 |
| TARGET | 36.31 | 32.23 | 34.89 | 34.48 | 33.33 | 39.23 | 28.32 | 38.23 | 24.76 | 35.45 | 22.85 | 38.25 | 20.71 | 42.23 | 19.25 | 45.23 |
| LANDER | 45.92 | 37.23 | 44.51 | 39.47 | 41.63 | 39.23 | 37.94 | 38.18 | 31.29 | 46.02 | 31.07 | 50.67 | 26.93 | 50.34 | 21.95 | 42.65 |
| CAN | **48.16** | **33.26** | **45.93** | **34.61** | **42.16** | **36.14** | **39.88** | **32.21** | **34.41** | **32.52** | **32.59** | **32.93** | **28.06** | **33.42** | **23.60** | **26.11** |
| LANDER | 43.99 | 40.71 | 43.46 | 41.70 | 40.85 | 41.49 | 37.65 | 38.89 | 30.25 | 47.83 | 28.82 | 53.64 | 25.46 | 52.65 | 20.37 | 44.26 |
| CAN | **46.16** | **35.01** | **44.82** | **27.80** | **42.35** | **37.56** | **38.91** | **35.10** | **33.96** | **33.43** | **31.52** | **34.18** | **27.32** | **28.72** | **22.09** | **31.75** |
| LANDER | 41.56 | 44.72 | 40.90 | 45.94 | 38.30 | 45.70 | 37.06 | 41.17 | 29.96 | 52.46 | 27.39 | 55.93 | 23.23 | 53.18 | 19.86 | 48.54 |
| CAN | **46.15** | **32.42** | **43.15** | **40.22** | **41.56** | **30.44** | **37.14** | **37.32** | **33.71** | **36.67** | **30.66** | **36.54** | **25.76** | **30.05** | **21.31** | **32.90** |
| LANDER | 36.99 | 51.87 | 36.41 | 52.74 | 35.63 | 49.80 | 33.34 | 46.59 | 27.00 | 56.70 | 23.86 | 46.71 | 22.16 | 55.83 | 19.00 | 52.60 |
| CAN | **44.92** | **29.51** | **43.64** | **34.34** | **41.04** | **34.65** | **35.38** | **43.63** | **31.83** | **37.34** | **27.13** | **42.75** | **24.99** | **32.87** | **21.07** | **33.91** |

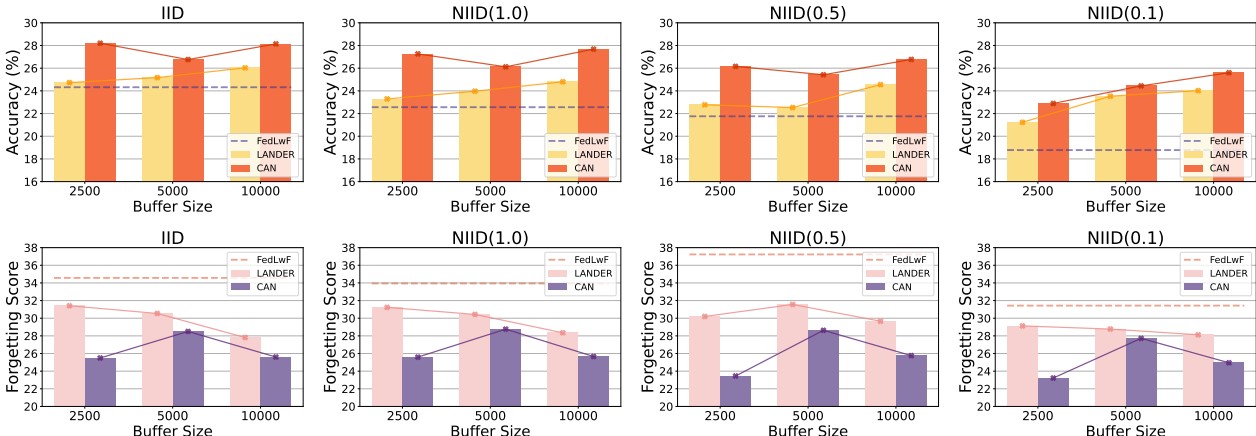

*Figure 4.* **Results on TinyImageNet** across 5 tasks under various data settings (IID, NIID 1.0, NIID 0.5, and NIID 0.1). The first row shows accuracy (↑), while the second row displays the forgetting score (↓). We compare CAN with LANDER at different buffer sizes (2,500, 5,000, and 10,000 samples) and include FedLwF as a baseline (the dashed line). CAN consistently outperforms other methods, demonstrating superior accuracy and lower forgetting. Please see Sec. 5.3 for a detailed discussion.

costs. This demonstrates CAN's efficiency, effectively leveraging client-specific expertise and adaptive buffer allocation to maintain robust performance.

### 5.3. Results on More Complicated Settings

As shown in Fig. 4, CAN consistently achieves state-of-the-art performance on TinyImageNet (Le & Yang, 2015), particularly in non-IID settings. It outperforms both LANDER (Tran et al., 2024) and FedLwF (Li & Hoiem, 2017), demonstrating robustness in handling diverse data distributions. Notably, we observe a "V" phenomenon with CAN: as buffer size increases, accuracy initially dips before rising, while forgetting follows an inverted "V" pattern, suggesting a trade-off between data quality and quantity. For LANDER, larger buffers compensate for lower data quality, whereas CAN's high-quality synthetic data benefits less from simply adding more samples. This suggests future improvements should focus on further enhancing data quality to better leverage larger buffers and optimize FCL performance in federated settings.

*Table 2.* **Average accuracy trajectory** measured after the completion of each task on **ImageNet100** over a sequence of 5 tasks. CAN consistently outperforms LANDER across all tasks and non-IID settings, demonstrating its effectiveness in maintaining higher accuracy throughout the sequence. Please refer to Sec. 5.3.

| Settings | Method | Task 2 | Task 3 | Task 4 | Task 5 | Avg |
|---|---|---|---|---|---|---|
| NIID(0.5) | LANDER | 40.40 | 31.30 | 23.92 | 17.48 | 28.28 |
| | CAN (Ours) | **50.15** | **40.87** | **32.70** | **26.52** | **37.56** |
| NIID(0.1) | LANDER | 26.20 | 15.90 | 10.75 | 7.02 | 14.97 |
| | CAN (Ours) | **40.90** | **31.00** | **23.30** | **17.50** | **28.18** |

*Table 3.* **Retention of final accuracy on the first task** after the completion of all tasks on CIFAR100. (±) indicate the difference compared to CAN with 500 buffer size. CAN consistently outperforms LANDER across all buffer sizes, even when LANDER uses twice as much data. Please refer to Sec. 5.3 for further analysis.

| Method | IID | NIID(1.0) | NIID(0.5) | NIID(0.1) |
|---|---|---|---|---|
| LANDER *500 | 18.20 (-20.20) | 14.65 (-19.40) | 14.55 (-16.00) | 15.75 (-6.00) |
| LANDER *750 | 24.65 (-13.75) | 21.30 (-12.75) | 20.35 (-10.20) | 20.20 (-1.55) |
| LANDER *1000 | 29.70 (-8.70) | 26.95 (-7.10) | 23.25 (-7.30) | 21.80 (+0.05) |
| LANDER *1250 | 33.05 (-5.35) | 28.20 (-5.85) | 24.35 (-6.20) | 22.25 (+0.50) |
| CAN *500 | **38.40** | **34.05** | **30.55** | **21.75** |

**Accuracy Trajectory on ImageNet.** To further evaluate the robustness of our method, we investigate its accuracy trajectory on the more challenging ImageNet100 (Russakovsky et al., 2015) dataset over a sequence of five consecutive tasks. As presented in Tab. 2, CAN consistently outperforms LANDER (Tran et al., 2024) across all tasks, exhibiting a substantial improvement in average accuracy. This performance gap becomes even more pronounced under the more challenging NIID(0.1) setting, where CAN achieves an average accuracy of 28.18%, significantly exceeding LANDER's 14.97%. These results highlight the effectiveness of our client-centric approach in maintaining strong performance across diverse learning conditions.

**Retention of Initial Task Accuracy.** Tab. 3 presents the final accuracy on the initial task, offering a targeted evaluation of how well FCL methods retain early knowledge. Unlike the average accuracy across all tasks, which may obscure the forgetting patterns of individual tasks, this measure directly reflects a model's ability to preserve knowledge. CAN consistently outperforms LANDER, even when using smaller buffer sizes. Notably, CAN achieves superior retention of initial task accuracy even as LANDER utilizes buffer sizes up to twice as large, underscoring CAN's efficiency and effectiveness in mitigating forgetting.

### 5.4. Ablation Studies

As shown in Fig. 5, we analyze the impact of removing key components from our method. The variant that excludes both client expertise and adaptive replay exhibits a significant performance decline. Specifically, without client expertise, data generation depends solely on server knowledge, while the absence of adaptive replay results in

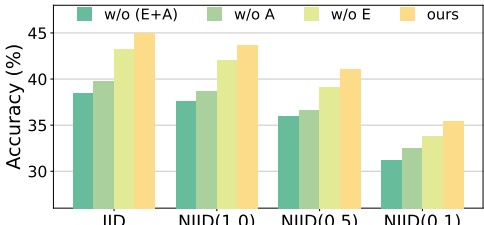

*Figure 5.* **Ablation study** exploring the importance of client expertise in data synthesis (E) and adaptive replay (A) across different settings, as discussed in Sec. 5.4, highlighting their critical roles.

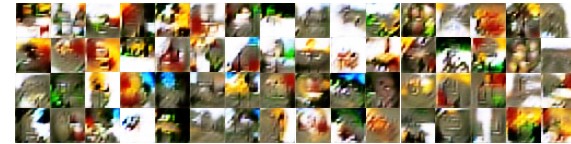

*Figure 6.* **Visualization** of the synthetic data generated by CAN. The synthetic data represents features in the form of meta data, abstracting class distributions without resembling real samples, ensuring privacy preservation. Details refer to Sec. 5.5.

uniform buffer allocation across tasks. These findings highlight two critical factors for replay-based FCL: 1) the quality of synthetic data and 2) the effectiveness of buffer allocation. Notably, the removal of adaptive replay leads to the most substantial accuracy drop, underscoring the necessity of tailored replay strategies to mitigate client-specific forgetting patterns. Furthermore, the exclusion of client expertise also degrades performance, demonstrating its essential role in enhancing synthetic data quality.

### 5.5. Privacy Analysis of CAN

CAN builds upon the generative replay FCL framework, similar to prior works like TARGET (Zhang et al., 2023) and LANDER (Tran et al., 2024). It leverages client models on the server side for knowledge transfer, synthesizing high-quality representations without direct access to original data and without introducing additional privacy concerns. As shown in Fig. 6, the synthetic data exhibits distinct visual differences from real data, serving as abstract representations of class distributions. Throughout the entire process, we ensure that each client's private data remains local and inaccessible to others. The synthetic data distributed for replay serves only as meta-level representations, effectively capturing features while remaining free from any identifiable real-data characteristics, thereby safeguarding privacy.

## 6. Conclusion

In this paper, we introduce CAN, a client-centric approach that enhances synthetic data quality and mitigates forgetting through adaptive replay. Unlike prior methods, CAN tailors replay strategies to client-specific needs, improving knowledge retention in FCL. Experiments show that CAN

outperforms existing baselines, even with smaller buffers, highlighting the effectiveness of client-aware strategies in FCL and paving the way for future research.

## Acknowledgement

This work is supported by National Natural Science Foundation of China under Grant (62361166629, 62176188), and the National Key Research and Development Program of China (2024YFC3308400). The supercomputing system at the Supercomputing Center of Wuhan University supported the numerical calculations in this paper.

## Impact Statement

This paper presents work whose goal is to advance the field of Machine Learning. There are many potential societal consequences of our work, none which we feel must be specifically highlighted here.

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

# A. Experiment Details

## A.1. Evaluation Metrics

We employ two metrics to assess the performance of our method in the federated continual learning setting.

**Average Accuracy ($\mathcal{A}$)** : This metric evaluates the model's classification performance on the test set after completing the final task, reflecting its ability to learn new tasks while retaining previously acquired knowledge. It is computed as:

$$\mathcal{A} = \frac{1}{|D_T|} \sum_{(x,y) \in D_T} \mathbb{I}(\hat{y} = y) \tag{10}$$

where $D_T$ represents the test dataset of the overall tasks, and $\mathbb{I}(\cdot)$ is the indicator function that returns 1 if the predicted label $\hat{y}$ matches the ground truth $y$, and 0 otherwise.

**Average Forgetting Score ($\mathcal{F}$)** : This metric quantifies the extent of knowledge degradation across previously learned tasks due to the introduction of new tasks. We first compute the maximum accuracy achieved for each task throughout training and compare it with the final accuracy after training all tasks. The forgetting rate for task $t$ is defined as:

$$\mathcal{F}_t = \max_{i \leq t} \mathcal{A}_{i,t} - \mathcal{A}_{T,t} \tag{11}$$

where $A_{i,t}$ is the accuracy of task $t$ after training on task $i$, and $A_{T,t}$ is the final accuracy on task $t$ after training on all $T$ tasks. The **Average Forgetting Rate ($\mathcal{F}$)** is then computed as:

$$\mathcal{F} = \frac{1}{T-1} \sum_{t=1}^{T-1} \mathcal{F}_t \tag{12}$$

## A.2. Generator Architecture and Optimization

To ensure a fair comparison under consistent conditions, we adopt the same generator architecture as LANDER (Tran et al., 2024) (as shown in Tab. 4 and Tab. 5). To optimize the generator, Adam optimizer is employed with a learning rate of 2e-3, ensuring stable convergence. The synthetic batch size is adjusted based on dataset complexity and hardware constraints, set to 256 for CIFAR-100 and Tiny-ImageNet, and 128 for ImageNet100 to maintain GPU memory usage below 24GB (NVIDIA RTX 4090). This design ensures that the generator remains scalable and efficient across diverse tasks, contributing to the robustness of our method in challenging settings.

*Table 4.* Generator Architecture for CIFAR-100 and TinyImageNet.

| Layers | Output |
|---|---|
| Input | 256 |
| Linear, BatchNorm1D, Reshape | $128 \times h/4 \times w/4$ |
| SpectralNorm (Conv ($3 \times 3$)), BatchNorm2D, LeakyReLU | $128 \times h/4 \times w/4$ |
| UpSample ($2\times$) | $128 \times h/2 \times w/2$ |
| SpectralNorm (Conv ($3 \times 3$)), BatchNorm2D, LeakyReLU | $64 \times h/2 \times w/2$ |
| UpSample ($2\times$) | $64 \times h \times w$ |
| SpectralNorm (Conv ($3 \times 3$)), Sigmoid, BatchNorm2D | $3 \times h \times w$ |

*Table 5.* Generator Architecture for ImageNet100.

| Layers | Output |
|---|---|
| Input | 256 |
| Linear, BatchNorm1D, Reshape | $128 \times h/16 \times w/16$ |
| SpectralNorm (Conv $(3 \times 3)$), BatchNorm2D, LeakyReLU | $128 \times h/16 \times w/16$ |
| UpSample $(2\times)$ | $128 \times h/8 \times w/8$ |
| SpectralNorm (Conv $(3 \times 3)$), BatchNorm2D, LeakyReLU | $128 \times h/8 \times w/8$ |
| UpSample $(2\times)$ | $128 \times h/4 \times w/4$ |
| SpectralNorm (Conv $(3 \times 3)$), BatchNorm2D, LeakyReLU | $64 \times h/4 \times w/4$ |
| UpSample $(2\times)$ | $64 \times h/2 \times w/2$ |
| SpectralNorm (Conv $(3 \times 3)$), BatchNorm2D, LeakyReLU | $64 \times h/2 \times w/2$ |
| UpSample $(2\times)$ | $64 \times h \times w$ |
| SpectralNorm (Conv $(3 \times 3)$), Sigmoid, BatchNorm2D | $3 \times h \times w$ |

## A.3. Hyperparameter Choice

### A.3.1. DEFAULT HYPERPARAMETER

To clarify the experimental setup, we provide the key hyperparameter configurations in the table below. These settings were consistently applied across all experiments unless otherwise specified.

*Table 6.* Training parameters.

| Hyperparameter | Value |
|---|---|
| Communication Rounds | 100 |
| Clients Numbers | 5 |
| Local Epochs | 2 |
| Local Batch Size | 128 |
| Synthesis Batch Size | 256 |

### A.3.2. BALANCING $\mathcal{L}_{nav}$ AND $\mathcal{L}_{gap}$

We analyze the effect of varying $\lambda$ in Eq. (5) to balance $\mathcal{L}_{\text{nav}}$ and $\mathcal{L}_{\text{gap}}$. As shown in Tab. 7, the results indicate that placing greater emphasis on $\mathcal{L}_{\text{gap}}$ (higher $\lambda$ values) leads to improved performance. This trend suggests that while prominent features are relatively easy to capture through $\mathcal{L}_{\text{nav}}$, the finer, more nuanced details—although harder to learn—are also crucial in real-world scenarios and thus warrant additional focus. The stronger weighting on $\mathcal{L}_{\text{gap}}$ enables the generator to refine and diversify synthetic data by capturing these subtle characteristics. This finding further supports the importance of introducing the gap loss $\mathcal{L}_{\text{gap}}$ to achieve a richer and more comprehensive data.

*Table 7.* Effect of varying the value of $\lambda$ on accuracy, as defined in the joint loss function $\mathcal{L}$ in Eq. (5). Increasing $\lambda$ (placing more emphasis on $\mathcal{L}_{\text{gap}}$) leads to better performance.

| $\lambda$ | 0.33 | 0.5 | 0.67 |
|---|---|---|---|
| Acc ($\mathcal{A}$) | 41.01 | 42.03 | **43.25** |

### A.3.3. ITERATIONS DURING PROFILING STAGE

In Tab. 8, we find that setting $\tau$ to 20 consistently yields either the best or second-best accuracy across different buffer sizes, making it an effective choice for balancing efficient training and capturing the impact of new tasks. Additionally, we observe that the accuracy improves when the loss stabilizes around 150 during training. For simplicity and consistency, we use $\tau = 20$ as the basis for measuring forgetting patterns in our experiments.

*Table 8.* Effect of varying the number of iterations $\tau$ during the profiling stage across different buffer sizes, highlighting the **best** and second-best performance.

| Size | $\tau$ | 5 | 10 | 15 | 20 | 25 |
|---|---|---|---|---|---|---|
| 500 | Acc ($\mathcal{A}$) | 37.12 | 37.62 | 41.44 | **42.33** | 41.22 |
| | Loss ($\mathcal{L}$) | 162.75 | 160.76 | 152.33 | 147.97 | 146.80 |
| 750 | Acc ($\mathcal{A}$) | 41.53 | 40.86 | 41.56 | 42.36 | **42.58** |
| | Loss ($\mathcal{L}$) | 166.93 | 162.92 | 161.03 | 155.80 | 151.87 |
| 1000 | Acc ($\mathcal{A}$) | 41.78 | 42.06 | **42.35** | 42.27 | 41.96 |
| | Loss ($\mathcal{L}$) | 160.31 | 154.29 | 153.03 | 150.35 | 147.54 |
| 1250 | Acc ($\mathcal{A}$) | 40.78 | 41.23 | 41.46 | **42.34** | 40.02 |
| | Loss ($\mathcal{L}$) | 161.77 | 155.32 | 151.81 | 149.45 | 147.17 |

### A.3.4. PARAMETER SELECTION ON LOCAL TRAINING

Following Gao et al. (Gao et al., 2022b), we adopt an adaptive strategy for setting the scaling factors $\alpha_{new}^t$ and $\alpha_{pre}^t$, which balances the learning of new and previously encountered knowledge. As the ratio of previous classes to new classes increases, the difficulty of preserving past knowledge also grows. To address this, the scaling factors are defined as:

$$\alpha_{new}^t = \frac{1 + 1/\kappa}{\delta} \alpha_{cur}, \quad \alpha_{pre}^t = \kappa \delta \alpha_{pre}, \tag{13}$$

where

$$\kappa = \log_2 \left( \frac{|\mathcal{Y}^t|}{2} + 1 \right), \quad \delta = \sqrt{\frac{|\mathcal{Y}^{1:t-1}|}{|\mathcal{Y}^t|}}, \tag{14}$$

and $|\mathcal{Y}^t|$ represents the number of classes in task $t$. Here, $\alpha_{new}$ and $\alpha_{pre}$ are the base scaling factors.

This formulation ensures a dynamic adjustment of the importance weights, mitigating the challenges posed by an increasing number of prior classes. The logarithmic function in $\kappa$ provides a smooth scaling mechanism, while the square-root term in $\delta$ prevents excessive emphasis on earlier tasks. For more details, we refer readers to Gao et al. (Gao et al., 2022b).

## B. Further Comparison and analysis

Generative replay is a key technique in FCL to mitigate catastrophic forgetting. Two prominent approaches to replay-based methods are:

- Diffusion-Based Generative Replay: Diffusion models have recently gained attention due to their ability to generate high-quality synthetic samples for replay. These models iteratively refine noise into structured data distributions, enabling data-free continual learning.

- Data-Free Knowledge Distillation (DFKD): Our approach leverages data-free knowledge distillation, which bypasses explicit data synthesis by directly distilling knowledge from previous models into new tasks. This eliminates the need for storing synthetic samples while still retaining knowledge through logit-based or feature-based distillation.

### B.1. Performance Analyze

In this section, we compare CAN with two state-of-the-art methods that utilize pretrained models for generative replay: DDDR (Liang et al., 2025), which leverages a pretrained diffusion model, and LANDER (Tran et al., 2024), which incorporates pretrained language models. Unlike these approaches, CAN does not rely on any external pretrained models but instead fully exploits the role of clients to achieve effective generative replay.

*Table 9.* Performance comparison on CIFAR-100 with DDDR and LANDER across different buffer sizes and data heterogeneity levels. Evaluated with Average Accuracy ($\mathcal{A}$).

| Buffer Size | Methods | IID | NIID(1.0) | NIID(0.5) | NIID(0.1) |
|---|---|---|---|---|---|
| 500 | DDDR | 44.21 | 41.82 | 40.06 | **36.93** |
| | LANDER | 36.99(-7.22) | 36.41(-5.41) | 35.63(-4.43) | 33.34(–3.59) |
| | CAN | **44.92**(+0.71) | **43.64**(+1.82) | **41.04**(+0.98) | 35.38(-1.55) |
| 750 | DDDR | 45.30 | **44.10** | 41.43 | **37.80** |
| | LANDER | 41.56(-3.74) | 40.90(-3.20) | 38.30(-3.13) | 37.06(-0.74) |
| | CAN | **46.15**(+0.85) | 43.15(-0.95) | **41.56**(+0.13) | 37.14(-0.66) |
| 1000 | DDDR | **46.81** | 44.51 | 41.11 | 38.55 |
| | LANDER | 43.99(-2.82) | 43.46(-1.05) | 40.85(-0.26) | 37.65(-0.90) |
| | CAN | 46.16(-0.65) | **44.82**(+0.31) | **42.35**(+1.24) | **38.91**(+0.35) |
| 1250 | DDDR | 46.50 | 45.20 | **43.04** | 39.06 |
| | LANDER | 45.92(-0.58) | 44.51(-0.69) | 41.63(1.41) | 37.94(-1.12) |
| | CAN | **48.16**(+1.66) | **45.93**(+0.73) | 42.16(-0.88) | **39.88**(+0.82) |

As shown in Tab. 9, DDDR achieves strong performance due to the superior generative capabilities of pretrained diffusion models, which enhance replay effectiveness. However, our proposed method, CAN, matches or even surpasses DDDR across various settings—all without relying on any pretrained models. Specifically, CAN consistently outperforms LANDER and achieves results comparable to DDDR, even under the most challenging NIID(0.1) scenario. This demonstrates that the strategic use of client-side knowledge alone is sufficient to bridge the performance gap between diffusion-based and traditional DFKD approaches, eliminating the need for computationally expensive pretrained models.

These results highlight the effectiveness of client-aware generative replay, showing that using pretrained models is not a prerequisite for strong performance in federated continual learning. Instead, CAN effectively leverages local client expertise to enhance knowledge retention and mitigate forgetting, achieving state-of-the-art results with significantly lower computational overhead.

## B.2. Cost Analysis

We analyze both the computational overhead (synthesis and local replay costs) and the storage requirements (buffer size and model storage) of CAN, comparing it with DDDR and LANDER.

### B.2.1. COMPUTATIONAL COST

**Synthesis Overhead.** DDDR relies on a pretrained diffusion model, which requires multiple iterative steps to produce high-quality synthetic samples. This multi-step process significantly increases inference time and computational complexity. In contrast, both CAN and LANDER are DFKD-based approaches that generate synthetic samples in a single step, making them substantially more efficient in terms of synthesis time and computational cost.

**Local Replay Overhead.** Local replay introduces additional computation during training. By maintaining a smaller replay buffer, CAN reduces the number of samples involved in this phase, resulting in shorter replay times and lower computational load. In comparison, LANDER and DDDR utilize larger buffers, prolonging the replay phase and increasing the overall training cost. Hence, replaying fewer samples allows CAN to accelerate local training and enhance efficiency in federated continual learning.

**GPU Memory Consumption.** DDDR must load a large pretrained diffusion model, which consumes considerable GPU memory due to its high parameter count and multi-step generation process. LANDER, reliant on pretrained language models, also incurs a sizable memory footprint. In contrast, CAN operates without any pretrained models, reducing GPU memory consumption and making it a practical option for resource-limited federated settings.

B.2.2. STORAGE COST

**Buffer Size Requirements.** CAN achieves competitive performance with a substantially smaller buffer (see Table 9), reducing storage needs while maintaining high accuracy. LANDER struggles to retain knowledge effectively due to lower-quality generated data and the absence of an adaptive buffer strategy, forcing it to rely on a larger replay buffer and incurring higher storage costs, yet still underperforming relative to CAN. Although DDDR benefits from the strong generative capabilities of a pretrained diffusion model, it lacks an adaptive replay strategy. As a result, despite producing high-quality synthetic samples, its performance remains on par with or slightly below CAN in most scenarios, illustrating that superior data quality alone is insufficient without a client-adaptive replay mechanism.

**Pretrained Model Storage Overhead.** DDDR demands substantial storage for a pretrained diffusion model, which can range from hundreds of megabytes to several gigabytes, depending on the architecture. LANDER also incurs notable storage costs due to its reliance on pretrained language models, which can occupy multiple gigabytes. CAN eliminates these overheads entirely by avoiding external pretrained models, drastically reducing storage requirements and enhancing scalability.

