# OpenReview forum: "CAN: Leveraging Clients As Navigators for Generative Replay in Federated Continual Learning"
_ICML.cc/2025/Conference — ICML 2025 poster_

### Official Review · Reviewer_TT97 · 2025-03-04

**Overall Recommendation:** 3

**Summary:**

This paper introduces Clients as Navigators (CAN) method, which tackling catastrophic forgetting in FCL tasks due to heterogeneous client data. CAN introduces a novel Generative Replay strategy that different from existing methods by selecting teaching clients for generator training and using an adaptive data distillation process to combat forgetting, dynamically adjusting to the client’s needs. The author responses match with my expectation ## update after rebuttal

**Claims And Evidence:**

Yes

**Essential References Not Discussed:**

None

**Experimental Designs Or Analyses:**

Yes, the experimental design is comprehensive, and the authors effectively validate the method’s effectiveness through extensive experiments.

**Methods And Evaluation Criteria:**

Yes. The methods are well evaluated.

**Other Comments Or Suggestions:**

None

**Other Strengths And Weaknesses:**

Strengths
1. This paper is well-motivated and provides a thorough analysis of why client's expertise knowledge is critical. The work has potential value across multiple related fields.
2. The paper's figure is color-coordinated and well-constructed, making it very easy for me to understand the content.
3. The experiments are thorough, and the experimental results sufficiently demonstrate the effectiveness of the method.


Weaknesses
1.	The challenges associated with FCL mentioned in the article are very interesting. To address these challenges, the authors proposed CAN, which has been shown to perform well through extensive experimentation. However, the improvements in methodology that CAN seems to be not substantial compared to the baseline LANDER. I am quite curious about the specific methodological improvements.
2.	Although the author has already provided the settings for some of the hyperparameters in the experiments, I would like to know about the settings for other important parameters, such as the number of communication rounds, the number of clients, and so on.

**Questions For Authors:**

I am curious whether the buffer size affects the allocation algorithm mentioned in the article. When the buffer size is small enough, is there a possibility that the accuracy of some labels is too low to allow for proper allocation?

**Relation To Broader Scientific Literature:**

The client-centric approach proposed by the authors offers valuable insights for the Federated Continual Learning (FCL) field, as previous research has rarely explored this perspective.

**Theoretical Claims:**

No specific theoretical claims are made.

---

> ### Author Rebuttal · Authors · 2025-03-29
>
> Dear Reviewer TT97:
>
> We appreciate your engagement with our work and the thoughtful observations you made. We aim to address your concerns in our detailed responses below, hoping to provide clarity and demonstrate the effectiveness of our proposed approach.
>
> ### Response to Weaknesses
>
> > **W1:** The challenges associated with FCL mentioned in the article are very interesting. To address these challenges, the authors proposed CAN, which has been shown to perform well through extensive experimentation. However, the improvements in methodology that CAN seems to be not substantial compared to the baseline LANDER. I am quite curious about the specific methodological improvements.
>
> **Our Response:** Thank you for your interest in the methodological contributions of CAN. A similar concern regarding the distinction between CAN and prior approaches such as LANDER was raised by Reviewer aG9E. To avoid redundancy, we have provided a detailed explanation in our response to Reviewer aG9E, outlining the key innovations of CAN in terms of client-side expert supervision and adaptive buffer replay based on forgetting patterns. We kindly refer you to that response for the complete clarification.
>
> > **W2:** Although the author has already provided the settings for some of the hyperparameters in the experiments, I would like to know about the settings for other important parameters, such as the number of communication rounds, the number of clients, and so on.
>
> **Our Response:** Thank you for your suggestion. To clarify the experimental setup, we provide the key hyperparameter configurations in the table below. These settings were consistently applied across all experiments unless otherwise specified.
>
> |   Hyperparameters    | Value |
> | :------------------: | :---: |
> | Communication Rounds |  100  |
> |   Clients Numbers    |   5   |
> |     Local Epochs     |   2   |
> |   Local Batch Size   |  128  |
> | Synthesis Batch Size |  256  |
>
>
>
> ### Response to Questions
>
> > **Q1:** I am curious whether the buffer size affects the allocation algorithm mentioned in the article. When the buffer size is small enough, is there a possibility that the accuracy of some labels is too low to allow for proper allocation?
>
> **Our Response:** Thank you for raising this important question. We agree that in scenarios with small buffer sizes, the accuracy on certain labels or tasks might be very low, which could potentially lead to unstable or skewed buffer allocations. To mitigate this issue, we introduce a predefined lower-bound threshold $ \epsilon $ in the calculation of the forgetting weight (Equation 7). Specifically, we define:
> $$
> w_c^t = \frac{1}{\max(\text{Acc}_c^t, \epsilon)}
> $$
> This ensures that when the classification accuracy $ \text{Acc}_c^t $ is too low, the forgetting weight is capped to avoid extreme values. By preventing division by near-zero accuracies, this threshold stabilizes the allocation algorithm and ensures a reasonable distribution of buffer space even under small-buffer regimes. The value of $ \epsilon $ is empirically chosen (set to 0.1 in our experiments) and does not require tuning across datasets.

---

### Official Review · Reviewer_z1gv · 2025-03-05

**Overall Recommendation:** 3

**Summary:**

CAN addresses the challenges imposed by non-IID data in Federated Continual Learning (FCL) and introduces a novel approach that leverages the specialized expertise of individual clients. By employing Expert-Driven Data Synthesis, CAN enhances the quality and representativeness of the generated data, ensuring effective retention of client-specific knowledge. Moreover, the integration of Adaptive Replay optimizes the utilization of synthesized data, improving memory efficiency and model stability over time. Collectively, these components enable CAN to improve continual learning performance within the Federated Learning framework, effectively addressing key issues in knowledge retention and adaptation across heterogeneous clients.

**Claims And Evidence:**

Yes

**Essential References Not Discussed:**

None

**Experimental Designs Or Analyses:**

Experiments are sound，extensive experiments across three datasets (CIFAR100, TinyImagenet, ImageNet100) with varying non-IID degrees.

**Methods And Evaluation Criteria:**

The methods are logically structured, and the proposed approach effectively mitigates the problem of catastrophic forgetting in federated continual learning.

**Other Comments Or Suggestions:**

It could be worth exploring how to apply such methods in Diffusion-based approaches within FCL.

**Other Strengths And Weaknesses:**

Pros:
1. The article is thoroughly composed, featuring well-organized figures and tables that effectively support the main findings.
 2. The proposed approach is particularly compelling, as leveraging client-specific expertise offers a novel perspective and valuable insights for the research community.
3. Experimental results demonstrate that CAN achieves state-of-the-art (SOTA) performance across various datasets and task configurations.

Cons:
1. The reference model Π is described as being trained on synthetic data; however, its architecture and training parameters (e.g., number of epochs, optimizer details) are not provided, which complicates a complete understanding of this section.
2. Given that Federated Learning emphasizes communication efficiency, quantifying aspects such as the number of communication rounds and the computational cost associated with generative replay would enhance the study’s practical relevance. The paper currently provides only a descriptive account without experimental evidence to substantiate these claims.

**Questions For Authors:**

Please refer to Strengths And Weaknesses. No extra questions.

**Relation To Broader Scientific Literature:**

Builds on FCL works like TARGET and LANDER but uniquely integrates client expertise.

**Theoretical Claims:**

No theroretical claims.

---

> ### Author Rebuttal · Authors · 2025-03-29
>
> Dear Reviewer z1gv:
>
> Thank you for your encouraging remarks and the critical questions you posed. We have reflected thoroughly on your feedback and provide our detailed responses below to further clarify our method and contributions.
>
> ### Response to Weaknesses
>
> > **W1**: The reference model Π is described as being trained on synthetic data; however, its architecture and training parameters (e.g., number of epochs, optimizer details) are not provided, which complicates a complete understanding of this section.
>
> **Our Response:** Thank you for the suggestion. A similar question regarding the reference model $ \Pi $ was raised by Reviewer aG9E. To avoid redundancy, we have provided a detailed response under that review, including training configuration, architecture, and computational overhead. We kindly refer you to our response to Reviewer aG9E for the complete explanation.
>
> > **W2:** Given that Federated Learning emphasizes communication efficiency, quantifying aspects such as the number of communication rounds and the computational cost associated with generative replay would enhance the study’s practical relevance. The paper currently provides only a descriptive account without experimental evidence to substantiate these claims.
>
> **Our Response:** Thank you for the suggestion! All experiments in our paper are conducted under a consistent setting of 100 communication rounds. To support our claims on computational efficiency, we provide a detailed comparison with DDDR (a diffusion-based method) and LANDER. As shown in the table below, CAN reduces overall computation cost by over 75% compared to DDDR (97 vs. 406 minutes). This highlights the advantage of avoiding multi-step diffusion generation, which is computationally intensive and less practical in federated settings. Compared to LANDER, CAN incurs the same client-side training cost (57 minutes), ensuring fairness in edge device usage. Although CAN introduces slightly more server-side cost due to expert-guided processing, we argue that this overhead is minor and acceptable, especially given that most computation in FL occurs on the client side.
>
> |        | Local Training | Image Generation | Overall |
> | :----: | :------------: | :--------------: | :-----: |
> |  DDDR  |      186       |       220        |   406   |
> | LANDER |       57       |        28        |   85    |
> |  CAN   |       57       |        40        |   97    |
>
> ### Response to Suggestions
>
> > **S1:** It could be worth exploring how to apply such methods in Diffusion-based approaches within FCL.
>
> **Our Response:** Thank you for the thoughtful suggestion. Integrating Diffusion-based approaches into the FCL setting is indeed a valuable direction. Compared to GANs, diffusion models are known to generate higher-quality and more diverse samples, which could further improve replay effectiveness in non-IID settings. However, applying diffusion in a federated and continual learning context also brings new challenges, such as communication efficiency, and privacy preservation. We believe this is a promising extension and plan to explore how our expert-guided replay strategy could be adapted to this setting in future work.

---

### Official Review · Reviewer_czny · 2025-03-08

**Overall Recommendation:** 4

**Summary:**

The paper focusing on Federated Continual Learning (FCL) scenario, highlights two key observations: Client Expertise Superiority and Client Forgetting Variance. The study shifts attention from the server to the client and proposes using the unique capabilities of client-side knowledge to improve Generative Replay. This new perspective explores how Federated Learning can reduce forgetting in FCL tasks.

**Claims And Evidence:**

Yes, the claims made in this paper are supported by clear and convincing evidence.

**Essential References Not Discussed:**

None.

**Experimental Designs Or Analyses:**

The experimental design is well-structured, and the tests were conducted on three widely used datasets: CIFAR100, TinyImageNet, and ImageNet100. However, the study falls short in including experiments with broader generalizability，see details in Weaknesses.

**Methods And Evaluation Criteria:**

Yes, The proposed method “CAN” is practical and easy to follow, make sense for the problem or application at hand.

**Other Comments Or Suggestions:**

Typo: Table 3 should refer to Sec. 5.4, not Sec. 5.3.

**Other Strengths And Weaknesses:**

Strengths:
1. The paper is well-written and clearly organized, making it easy to understand.
2. Comparative experiments demonstrate that the proposed method achieves notable effectiveness in terms of computational cost and reduction of forgetting rates.
3. The proposed method does not compromise the privacy-preserving capabilities of the federated learning paradigm.
Weaknesses
Personally, I don’t see any big problems with the method described in the paper. What I’m really curious about are some of the experimental details.
1. Could more experiments be conducted under additional buffer sizes, such as 2560 and 5120? I’d like to see whether CAN can still maintain the same level of performance under larger buffer size conditions.
2. The paper does not mention the number of clients in experiments, which is a critical factor in federated learning. Could experiments be conducted in scenarios with a larger number of clients?
3. The "V phenomenon" mentioned in Sec. 5.3 suggests a trade-off between buffer size and data quality. The authors should discuss how to optimize this balance in practice.

**Questions For Authors:**

I was wondering that which dataset was used for the experiments in Table 3? It doesn’t seem to be mentioned in the paper. No other questions.

**Relation To Broader Scientific Literature:**

No.

**Theoretical Claims:**

This paper doesn’t include theoretical claims, just with necessary formulas.

---

> ### Author Rebuttal · Authors · 2025-03-29
>
> Dear Reviewer czny:
>
> We are truly grateful for your insightful review and the constructive feedback provided. Your suggestions helped us identify areas for clarification, and we respond to each point below with careful consideration.
>
> ### Response to Weaknesses
>
> > **W1:** Could more experiments be conducted under additional buffer sizes, such as 2560 and 5120? I’d like to see whether CAN can still maintain the same level of performance under larger buffer size conditions.
>
> **Our Response:** Thanks for your valuable insights!  Following your advice, we conducted additional experiments under larger buffer sizes (2560 and 5120). As shown in the table below, CAN consistently outperforms LANDER across all settings, including under larger buffer budgets. Notably, even when the buffer is doubled (5120), CAN maintains a clear performance advantage under both IID and non-IID scenarios. These results further confirm the scalability and robustness of our method across different buffer regimes.
>
> |           | LANDER(2560) | CAN(2560) | LANDER(5120) | CAN(5120) |
> | :-------: | :----------: | :-------: | :----------: | :-------: |
> |    IID    |    47.97     | **49.36** |    49.19     | **51.81** |
> | NIID(1.0) |    46.33     | **48.31** |    47.82     | **50.86** |
> | NIID(0.5) |     43.3     | **46.3**  |    45.43     | **47.72** |
> | NIID(0.1) |    40.01     | **40.98** |    42.14     | **43.72** |
>
> > **W2:** The paper does not mention the number of clients in experiments, which is a critical factor in federated learning. Could experiments be conducted in scenarios with a larger number of clients?
>
> **Our Response:** We appreciate the suggestion. In our main experiments, we used 5 clients by default. To evaluate scalability under more realistic federated settings, we conducted additional experiments with 10 and 15 clients. As shown in the table above, CAN consistently outperforms LANDER across all client counts, especially under non-IID scenarios. These results demonstrate that CAN maintains its effectiveness even as the number of clients increases, confirming its robustness in more complex and distributed environments.
>
> |           | LANDER (10) | CAN (10)  | LANDER(15) |  CAN(15)  |
> | :-------: | :---------: | :-------: | :--------: | :-------: |
> |    IID    |    45.53    | **46.39** |   45.32    | **45.68** |
> | NIID(1.0) |    43.00    | **44.64** |   41.79    | **42.65** |
> | NIID(0.5) |    41.53    | **42.81** |   40.02    | **40.25** |
> | NIID(0.1) |    27.89    | **28.86** |   22.73    | **24.28** |
>
> > **W3:** The "V phenomenon" mentioned in Sec. 5.3 suggests a trade-off between buffer size and data quality. The authors should discuss how to optimize this balance in practice.
>
> **Our Response:** We appreciate the reviewer’s insightful observation. This behavior arises from the interplay between buffer size and adaptive allocation efficiency. When the buffer is small, the system allocates limited replay slots more selectively to the most forgotten tasks, resulting in efficient knowledge retention. As the buffer increases, this selective effect weakens, and additional samples contribute less to mitigating forgetting, leading to a temporary dip in performance. In practice, we suggest starting from a moderate buffer budget and gradually increasing it while monitoring key metrics such as retention accuracy on early tasks and per-task replay effectiveness. If performance plateaus or degrades, it may indicate that further buffer increase offers diminishing returns. Such observations can guide the choice of buffer size under memory or communication constraints.
>
> ### Response to Suggestions
>
> > **S1:** Typo: Table 3 should refer to Sec. 5.4, not Sec. 5.3.
>
> **Our Response:** Thank you for pointing this out. We acknowledge the typo and will revise the reference from Sec. 5.3 to Sec. 5.4 in the final version of the paper.
>
> ### Response to Questions
>
> > **Q1:** I was wondering that which dataset was used for the experiments in Table 3? It doesn’t seem to be mentioned in the paper. No other questions.
>
> **Our Response:** Thanks for pointing this out! The experiments presented in Table 3 were conducted on the CIFAR-100 dataset. This table reports the final accuracy on the first task after the completion of all tasks, which serves as an important metric to evaluate catastrophic forgetting. We will clarify it in the final version of the paper.

---

### Official Review · Reviewer_aG9E · 2025-03-09

**Overall Recommendation:** 3

**Summary:**

This paper examines Federated Continual Learning, specifically exploring how client expertise aids in generating replay data and adjusting replay buffer sizes based on the forgetting variance among clients. The CAN approach first identifies accurate expert clients, utilizing their predictions to train the generator. By promoting discrepancies between the server model (S) and reference model (Π), it extracts fine-grained features. Additionally, the replay buffer is dynamically managed during training to minimize further forgetting.

**Claims And Evidence:**

Yes.

**Essential References Not Discussed:**

None

**Experimental Designs Or Analyses:**

The experimental results are comprehensive, with the author conducting various experiments under different settings to ensure thorough evaluation.

**Methods And Evaluation Criteria:**

Yes.

**Other Comments Or Suggestions:**

The **Expert-Driven Data Synthesis** section lacks a clear focus, making it difficult to grasp the key points. The authors need to improve the logical flow of this section.

**Other Strengths And Weaknesses:**

Strengths
1. The idea of the method is intuitive and easy to follow.
2. The authors provide a clear motivation and a well-structured discussion of related work.
3. The experiments conducted across different settings effectively demonstrate the strategy’s effectiveness and generalizability.

Weaknesses
1. The paper mentions that CAN is similar to TARGET [1] and LANDER [2]. However, as far as I know, both methods fall under the category of Data-Free Knowledge Distillation (DFKD). The authors should provide a more detailed distinction and explanation regarding how their approach differs from these methods.
2. The explanation of the Synthetic Data Initialization section is unclear, which raises concerns. The authors should provide a more detailed and precise clarification of this part.
3. The origin of the Reference Model is not clearly explained. Does it require training from scratch? What is the computational overhead? How many training iterations are needed, and how are the training parameters set? These aspects should be addressed in more detail.
[1] Zhang, J., Chen, C., Zhuang, W., and Lyu, L. Target: Federated class-continual learning via exemplar-free distillation. In CVPR, pp. 4782–4793, 2023.
[2] Tran, M.-T., Le, T., Le, X.-M., Harandi, M., and Phung, D. Text-enhanced data-free approach for federated class-incremental learning. In CVPR, pp. 23870–23880, 2024.

**Questions For Authors:**

Please refer to the Strengths and Weaknesses section for specific details.

**Relation To Broader Scientific Literature:**

None

**Theoretical Claims:**

The author provides multiple formulas to explain their method, and the theoretical claims are correct.

---

> ### Author Rebuttal · Authors · 2025-03-29
>
> Dear Reviewer aG9E:
>
> We greatly value the time and expertise you invested in reviewing our submission. Your feedback has been instrumental in helping us improve the clarity of our work. We address your comments in detail below.
>
> ### Response to Weaknesses
>
> > **W1:** The paper mentions that CAN is similar to TARGET and LANDER. However, as far as I know, both methods fall under the category of Data-Free Knowledge Distillation (DFKD). The authors should provide a more detailed distinction and explanation regarding how their approach differs from these methods.
>
> **Our Response:** We appreciate your point regarding the similarity between CAN and existing Data-Free Knowledge Distillation (DFKD) methods. While all three methods adopt a generative replay framework, CAN introduces two key innovations that fundamentally distinguish it from TARGET and LANDER:
>
> 1. **Client Expertise Superiority**: Unlike TARGET and LANDER, which rely solely on the global server model for guiding data synthesis, CAN explicitly identifies and leverages *client-side experts* to guide generation. Specifically, our *expertise map* selects the best-performing client for each class, whose model is then used to supervise the generator through an *Expert Navigation Loss*. This enables class-specific fine-grained guidance that cannot be captured by a centralized server alone.
> 2. **Client Forgetting Variance**: Prior methods use uniform replay buffers across clients, overlooking the heterogeneous forgetting behavior induced by non-IID data. In contrast, CAN profiles each client’s unique forgetting patterns and adapts the buffer allocation accordingly, improving both performance and efficiency.
>
> > **W2:** The explanation of the Synthetic Data Initialization section is unclear, which raises concerns. The authors should provide a more detailed and precise clarification of this part.
>
> **Our Response:**  Thank you for pointing this out. We agree that the Synthetic Data Initialization section could benefit from more detailed clarification. Below we provide a clearer description of the process.
>
> The initialization begins by sampling class-conditional latent codes $ z_y \sim \mathcal{N}(0, I) $, where $ y $ denotes the target class. These latent codes are passed through the generator $ G $ to obtain synthetic images $ x_{\text{syn}} = G(z_y) $. We then use the server model to perform inference on each $ x_{\text{syn}} $, obtaining predicted class probabilities. To ensure that the generated samples are aligned with their intended class labels, we compute an entropy-based loss between the prediction and the target label $ y $. This loss encourages the generator to produce more discriminative and class-consistent samples. Once the initial set of synthetic data is generated, we train the reference model $ \Pi $ from scratch on this data. The purpose of $ \Pi $ is to capture fine-grained features potentially missed by the server model. We then compare the predictions of $ \Pi $ and the server on synthetic samples and apply a KL divergence loss (described as $ \mathcal{L}_{\text{gap}} $) to encourage the generator to produce samples that expose the discrepancy between the two models. This results in more informative and transferable data for replay. We will revise Section 4.2.1 in the final version to clearly explain this.
>
> > **W3:** The origin of the Reference Model is not clearly explained. Does it require training from scratch? What is the computational overhead? How many training iterations are needed, and how are the training parameters set? These aspects should be addressed in more detail.
>
> **Our Response:** Thank you for the helpful comment! Please allow us to explain it. The purpose of the reference model $ \Pi $ is to guide the generator to produce more diverse samples. $ \Pi $  is trained from scratch on synthetic data and shares the same architecture as the server model (a ResNet-based classifier). This helps encourage the generation of informative, boundary-sensitive synthetic data, rather than overfitting to overly confident or less representative examples. We train it for 40 epochs using SGD with a learning rate of 0.1, momentum 0.9, and weight decay 0.0002. In terms of computational overhead, $ \Pi $ is only used during the training phase to guide the generator and is lightweight and isolated from client participation. It introduces less than 5% additional training time, and does not affect inference or deployment, thus incurring minimal cost in practice. These implementation details will be included in the appendix.
>
> ### Response to Suggestions
>
> > **S1:** The **Expert-Driven Data Synthesis** section lacks a clear focus, making it difficult to grasp the key points. The authors need to improve the logical flow of this section.
>
> **Our Response:** Thanks for your valuable suggestions to improve the presentation of our paper! We will revise the *Expert-Driven Data Synthesis* section to improve its clarity and logical flow in the final version.

---

### Decision · Program_Chairs · 2025-05-01

**Decision:**

Accept (poster)

**Comment:**

This paper discussed Generative Replay in Federated Continual Learning. After some discussion, the overall verdict was that this submission could be accepted in ICML-2025 while the concerns of reviewers should be addressed in the final version.